# SiDGen: Structure-informed Diffusion for Generative modeling of Ligands for Proteins

## Abstract

Designing ligands that are both chemically valid and structurally compatible with protein binding pockets is a key bottleneck in computational drug discovery. Existing approaches either ignore structural context or rely on expensive, memory-intensive encoding that limits throughput and scalability. We present SiDGen (Structure-informed Diffusion Generator), a protein-conditioned diffusion framework that integrates masked SMILES generation with lightweight folding-derived features for pocket awareness. To balance expressivity with efficiency, SiDGen supports two conditioning pathways: a streamlined mode that pools coarse structural signals from protein embeddings and a full mode that injects localized pairwise biases for stronger coupling. A coarse-stride folding mechanism with nearest-neighbor upsampling alleviates the quadratic memory costs of pair tensors, enabling training on realistic sequence lengths. Learning stability is maintained through in-loop chemical validity checks and an invalidity penalty, while large-scale training efficiency is restored via selective compilation, dataloader tuning, and gradient accumulation. In automated benchmarks, SiDGen generates ligands with high validity, uniqueness, and novelty, while achieving competitive performance in docking-based evaluations and maintaining reasonable molecular properties. These results demonstrate that SiDGen can deliver scalable, pocket-aware molecular design, providing a practical route to conditional generation for high-throughput drug discovery.

## 1 Introduction

Structure-based drug design (SBDD) aims to generate small molecules that bind specifically to target proteins, a fundamental challenge in computational drug discovery. Traditional approaches rely on virtual screening of large molecular libraries (Shoichet, 2004) or structure–activity relationship modeling (Cheng et al., 2007), but these methods are limited by finite chemical space exploration and dependence on existing molecular databases.

Recent advances in deep generative modeling have enabled *de novo* molecular design (Sanchez-Lengeling et al., 2017; Popova et al., 2018; Jin et al., 2018), but most methods operate unconditionally without target specificity (Brown et al., 2019; Polykovskiy et al., 2020). Protein-conditioned approaches address this by incorporating binding pocket information (Luo et al., 2022; Guan et al., 2023; Peng et al., 2022), but face a critical trade-off: Methods that use detailed 3D structural information achieve high expressivity at the cost of prohibitive computational overhead due to quadratic memory scaling with protein sequence length.

We present **SiDGen**-Structure-informed Diffusion Generator, a protein-conditioned diffusion framework that resolves the efficiency–expressivity trade-off through several key methodological developments. Our method integrates masked SMILES generation with lightweight folding-derived features, supporting dual conditioning pathways - a streamlined mode pooling coarse structural signals for efficiency and a full mode injecting localized pairwise biases for maximum expressivity. The core technical contribution is a **coarse-stride folding mechanism** that reduces quadratic memory complexity from $O(L^2)$ to $O((L/s)^2)$ through strategic down-sampling of protein features followed by nearest-neighbor up-sampling. This provides up to $16\times$ memory reduction for typical proteins while preserving essential structural information. We maintain training stability through **in-loop**

**validity checking** with **invalidity penalties** and **curriculum learning** that gradually increases denoising difficulty.

Our comprehensive evaluation demonstrates SiDGen's practical effectiveness:

- **High Generation Quality**: 100% validity, 88.75% uniqueness, and 100% novelty on MOSES benchmark
- **Strong Binding Performance**: Competitive ROC-AUC (0.819) and enrichment factors (EF@1% = 10–13) on DUD-E virtual screening
- **Accurate Affinity Prediction**: Pearson correlation of 0.6948 and RMSE of 1.0745 pKd on PDBBind dataset
- **Drug-like Properties**: Generated molecules exhibit reasonable molecular weight (421.57 Da), LogP (3.07), and synthetic accessibility (0.47)
- **Computational Efficiency**: Significant memory and compute savings enabling realistic sequence length processing

These results demonstrate that SiDGen delivers scalable, pocket-aware molecular design with competitive performance across multiple evaluation metrics, providing a practical route to conditional generation for high-throughput drug discovery applications where both quality and efficiency are paramount.

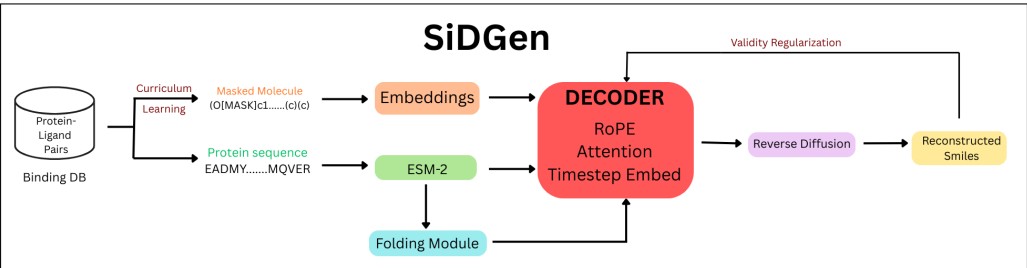

Figure 1: Overview of the SiDGen architecture.

## 2 RELATED WORK

### 2.1 MOLECULAR GENERATION METHODS

Early generative models for small molecules used variational autoencoders (VAEs) (Gómez-Bombarelli et al., 2018; Kusner et al., 2017) and generative adversarial networks (GANs) (De Cao & Kipf, 2018; Putin et al., 2018), facing challenges in chemical validity and diversity. Diffusion models (Hoogeboom et al., 2022; Wu et al., 2022; Vignac et al., 2022) have since improved both, treating molecule design as denoising from random noise.

TamGen (Wu et al., 2024) introduced a topology-aware graph generative approach that combines molecular graphs with hierarchical templates to optimize both structure and activity. TargetDiff (Zhang et al., 2023) leveraged diffusion to condition ligand generation directly on target protein pockets, achieving competitive affinity prediction and docking results using fine-grained 3D geometric features. REINVENT4 (Loeffler et al., 2024) employs reinforcement learning within generative models, using structure-based objectives and constraints to drive optimization in real drug design settings.

### 2.2 PROTEIN-CONDITIONED MOLECULAR GENERATION

The field of protein-conditioned molecular generation has evolved rapidly. Early approaches used simple sequence-based conditioning (Zhavoronkov et al., 2019; Born & Manica, 2021), but lacked structural awareness. More sophisticated methods have incorporated 3D structural information through various mechanisms.

Pocket2Mol (Luo et al., 2022) introduced a two-stage approach that first generates molecular graphs conditioned on binding pockets, then optimizes 3D coordinates. TargetDiff (Zhang et al., 2023) applied diffusion models to 3D molecular generation with pocket conditioning, achieving strong performance but requiring expensive 3D coordinate processing.

DiffSBDD (Schneuing et al., 2022) and related methods (Guan et al., 2023) have explored various architectural choices for incorporating structural information, but generally require full processing of protein coordinates and pairwise interactions.

Recent work has also explored fragment-based approaches (Powers et al., 2023), multi-modal conditioning (Chen et al., 2023), and reinforcement learning for optimization (Thomas et al., 2021). However, most of these approaches face the fundamental trade-off between structural awareness and computational efficiency.

### 2.3 STRUCTURAL CONDITIONING IN PROTEIN MODELS

The challenge of efficiently incorporating structural information is well-studied in protein modeling. AlphaFold (Jumper et al., 2021) and related methods use various approximation strategies, including MSA subsampling and coarse-graining techniques, to manage computational complexity.

In the context of molecular generation, some approaches have explored simplified structural representations (Zhang et al., 2021; Liu et al., 2022), but these often sacrifice important interaction details. Our coarse-stride mechanism is inspired by multi-scale processing techniques from computer vision (Ronneberger et al., 2015; He et al., 2016) but adapted specifically for the sequential nature of protein-ligand interactions.

## 3 METHODOLOGY

### 3.1 PROBLEM FORMULATION AND DATASET

We formulate protein-conditioned ligand generation as learning a conditional distribution $p(x \mid c)$, where $x$ is a ligand SMILES string and $c$ represents protein conditioning information. The conditioning information includes sequence embeddings $c_{\text{seq}}$ from ESM-2 (Lin et al., 2023) and structural embeddings $c_{\text{struct}}$ derived from our coarse-stride folding mechanism (Sec. 3.4).

We utilize the **BindingDB** dataset (bin, 2024), which contains 1,154,054 ligand-protein pairs. Each entry includes a ligand SMILES string, protein sequence, organism, PFAM structural annotation, cluster identifier, augmentation flag, and binding label. The dataset comprises 6,098 unique protein sequences with lengths ranging from 50 to 1,500 residues (mean length 514) and 785,740 unique SMILES strings. Protein sequences are represented using embeddings of dimension $d_{\text{seq}} = 1280$, obtained from ESM-2 (Lin et al., 2023), and SMILES strings are tokenized with a pretrained diffusion tokenizer.

For a protein sequence $P = \{p_1, \ldots, p_L\}$ of length $L$, we obtain sequence embeddings:

$$c_{\text{seq}} = \text{ESM-2}(P) \in \mathbb{R}^{L \times d_{\text{seq}}}. \tag{1}$$

Structural features are downsampled via coarse-stride folding (Sec. 3.4) to reduce memory from $O(L^2)$ to $O((L/s)^2)$.

### 3.2 MASKED DIFFUSION FOR SMILES GENERATION

We adapt masked diffusion language models (MDLMs) (Austin et al., 2021) to discrete SMILES token space. The forward corruption process replaces tokens with [MASK] according to a noise schedule:

$$q(x_t \mid x_0) = \prod_{i=1}^{|x_0|} \left[ \alpha_t \mathbf{1}_{x_t^{(i)} = x_0^{(i)}} + (1 - \alpha_t) \mathbf{1}_{x_t^{(i)} = [\text{MASK}]} \right], \tag{2}$$

where $x_t$ is the initial SMILES of the molecule which evolves over time $t$ with $\alpha_t = 1 - \sigma(t)$. Among many possibilities, we use:

$$\text{Sigmoid-Warped: } \sigma(t) = -\log(1 - \text{sigmoid}(\frac{t - \kappa}{\tau}) + \epsilon).$$

where $\kappa$, and $\tau$ are regularization constants and $\epsilon$ is for numeric safety.

The reverse process predicts original tokens:

$$p_\theta(x_0 \mid x_t, c) = \prod_{i=1}^{|x_t|} \text{softmax}(f_\theta(x_t, t, c)_i), \tag{3}$$

where $f_\theta$ is a Transformer decoder conditioned on protein features $c$.

### 3.3 DECODER ARCHITECTURE

The decoder is a multi-layer Transformer that maps a concatenation of (i) a learned *timestep token*, (ii) conditioning tokens, and (iii) ligand sequence embeddings into updated ligand representations.

**Inputs:**

1. $t \in \mathbb{R}^B$: scalar timesteps for each batch element $b = 1, \ldots, B$.
2. $\mathbf{S} \in \mathbb{R}^{B \times L \times H}$: ligand (SMILES) token embeddings of length $L$ and hidden dimension $H$.
3. $\mathbf{E} \in \mathbb{R}^{B \times L \times H}$: extra conditioning sequence of length $L$.

**Timestep Embeddings:** sinusoidal encodings are projected through a two-layer $MLP$ (Multi Layered Perceptron) with SiLU activations to form the timestep token:

$$\sigma(t) = \text{SiLU}\big(W_2\,\text{SiLU}(W_1\,\text{PE}(t))\big), \quad \sigma(t) \in \mathbb{R}^H. \tag{4}$$

**Decoder Input:** the final input sequence is the concatenation of the timestep token, optional conditioning, and ligand embeddings:

$$\mathbf{X} = [\,\sigma(t);\ \mathbf{E};\ \mathbf{S}\,] \in \mathbb{R}^{N \times H}, \quad N = 1 + M + L. \tag{5}$$

The timestep token occupies position 0 in $\mathbf{X}$.

**Rotary Position Embeddings (RoPE) (Su et al., 2024):**

$$\text{RoPE}(x_m, m) = R(m)x_m, \quad R(m) = \begin{pmatrix} \cos(m\theta) & -\sin(m\theta) \\ \sin(m\theta) & \cos(m\theta) \end{pmatrix}, \quad \theta = 10000^{-2i/d}, \tag{6}$$

where $x_m$ is the $m$-th token embedding and $d = H/h$ the per-head dimension.

**Attention:** each layer performs self-attention on SMILES tokens and cross-attention with protein embeddings:

$$\text{Self-Attention: } \text{MHA}(Q_{\text{lig}}, K_{\text{lig}}, V_{\text{lig}}), \tag{7}$$

$$\text{Cross-Attention: } \text{MHA}(Q_{\text{lig}}, K_{\text{prot}}, V_{\text{prot}}), \tag{8}$$

$$\text{Feed-Forward: } \text{FFN}(x) = \text{Linear}(\text{ReLU}(\text{Linear}(x))). \tag{9}$$

Here, $Q$, $K$, and $V$ denote the query, key, and value projections of the input embeddings for each attention layer. For self-attention, they are computed from the ligand sequence; for cross-attention, the keys and values come from the protein embeddings while queries come from the ligand tokens.

Thus, timestep information is propagated bidirectionally through the sequence via attention, enabling the decoder to condition ligand updates on both time $t$ and external context.

### 3.4 COARSE-STRIDE FOLDING OF STRUCTURAL FEATURES

To mitigate the quadratic scaling of structural features, we select a coarse subset of positions from the protein sequence using a stride $s$:

$$\mathcal{I} = \{0, s, 2s, \ldots, \lfloor L/s \rfloor \cdot s\}.$$

The downsampled single features are $\mathbf{s}_c \in \mathbb{R}^{B \times L_c \times C_{\text{single}}}$ and pair features $\mathbf{p}_c \in \mathbb{R}^{B \times L_c \times L_c \times C_{\text{pair}}}$, where $L_c = \lceil L/s \rceil$.

Formally, each coarse feature is obtained by selecting the corresponding positions in $\mathcal{I}$:

$$\mathbf{s}_c[b, k, :] = \mathbf{s}[b, \mathcal{I}_k, :], \tag{10}$$

$$\mathbf{p}_c[b, k, l, :] = \mathbf{p}[b, \mathcal{I}_k, \mathcal{I}_l, :], \tag{11}$$

for all batch indices $b$ and coarse positions $k, l \in \{0, \dots, L_c - 1\}$.

**Folding Operations**   On coarse features, we apply triangle attention and multiplication inspired by AlphaFold:

**Folding Operations**   On the coarse pair features $\mathbf{p}_c$, we apply triangle attention and triangle multiplication inspired by AlphaFold Jumper et al. (2021):

**Triangle Attention:**

$$\text{TriAttn}_{\text{start}}(\mathbf{p}_c): \quad \mathbf{p}_c[b, i, j] \leftarrow \text{Attention}_k(\mathbf{p}_c[b, i, k]), \tag{12}$$

$$\text{TriAttn}_{\text{end}}(\mathbf{p}_c): \quad \mathbf{p}_c[b, i, j] \leftarrow \text{Attention}_k(\mathbf{p}_c[b, k, j]). \tag{13}$$

**Triangle Multiplication:**

$$\text{TriMult}_{\text{out}}(\mathbf{p}_c): \quad \mathbf{p}_c[b, i, j] \leftarrow \sum_k \mathbf{p}_c[b, i, k] \odot \mathbf{p}_c[b, j, k], \tag{14}$$

$$\text{TriMult}_{\text{in}}(\mathbf{p}_c): \quad \mathbf{p}_c[b, i, j] \leftarrow \sum_k \mathbf{p}_c[b, k, i] \odot \mathbf{p}_c[b, k, j]. \tag{15}$$

**Upsampling Coarse Features**   Let $\mathbf{s}_c \in \mathbb{R}^{B \times L_c \times C_{\text{single}}}$ and $\mathbf{p}_c \in \mathbb{R}^{B \times L_c \times L_c \times C_{\text{pair}}}$ denote the downsampled single and pair features. We upsample them to the original sequence length $L$ using nearest-neighbor mapping:

$$\mathbf{s}_{\text{out}}[b, i, :] = \mathbf{s}_c[b, k, :], \quad k = \left\lfloor \frac{i}{s} \right\rfloor, \tag{16}$$

$$\mathbf{p}_{\text{out}}[b, i, j, :] = \mathbf{p}_c[b, k, l, :], \quad k = \left\lfloor \frac{i}{s} \right\rfloor, \quad l = \left\lfloor \frac{j}{s} \right\rfloor, \tag{17}$$

for all batch indices $b$ and positions $i, j \in \{0, \dots, L - 1\}$. This mapping ensures that each original position receives the structural information from its corresponding coarse block, resulting in block-wise constant features.

**Complexity Analysis**   The coarse-stride mechanism reduces memory and compute:

- Memory: $O(L^2) \rightarrow O((L/s)^2)$
- Compute: Triangle attention/multiplication scales similarly, giving $\sim s^2$ speedup
- Approximation quality depends on local homogeneity of structural signals

### 3.5   TRAINING ENHANCEMENTS

**Substitution Parameterization:** masked diffusion loss with substitution scaling:

$$\mathcal{L}_{\text{MDLM}} = -\sum_{i,t} \log p_\theta(x_0^{(i)} \mid x_t^{(i)}, c) \cdot \frac{d\sigma/dt}{\exp(\sigma) - 1}. \tag{18}$$

**Curriculum Learning**   To gradually increase the difficulty of the denoising task, we scale the timestep used for loss computation:

$$t_{\text{curriculum}} = \min\left(\max(\epsilon, \alpha_{\text{epoch}} t), 1\right), \qquad \alpha_{\text{epoch}} = \min\left(1, \frac{\text{epoch} + 1}{T_{\text{curriculum}}}\right), \tag{19}$$

where $t \in [0, 1]$ is the diffusion timestep, $\epsilon > 0$ is a small lower bound, epoch is the current training epoch, and $T_{\text{curriculum}}$ is the total number of curriculum epochs.

**Validity Regularization**   To encourage chemically valid SMILES, we add a penalty proportional to the fraction of invalid samples generated by the model:

$$\mathcal{L}_{\text{total}} = \mathcal{L}_{\text{MDLM}} + \lambda_{\text{valid}} \, \mathbb{E}_{x \sim p_\theta} \big[ \mathbf{1}_{x \text{ invalid}} \big], \tag{20}$$

where $\lambda_{\text{valid}}$ controls the strength of the penalty and $\mathbf{1}_{x \text{ invalid}}$ is an indicator function equal to 1 if $x$ is invalid and 0 otherwise.

## 4 RESULTS

### 4.1 MOLECULAR GENERATION QUALITY

Table 1 shows SiDGen's performance on the MOSES benchmark. Our model achieves 100% validity and 88.75% uniqueness, demonstrating reliable generation of chemically valid molecules. The internal diversity of 0.893 indicates good structural variety in generated molecules.

MOSES metrics (Validity, Uniqueness, Novelty, IntDiv) were computed on held-out generated samples using RDKit for SMILES canonicalization and validity checks. FCD (Fréchet ChemNet Distance) (Preuer et al., 2018) uses ChemNet embeddings; internal diversity and Tanimoto-based statistics use Morgan fingerprints (radius=2, n_Bits=2048).

Table 1: MOSES Benchmark Results

| Method | Validity | Uniqueness | Novelty | FCD $\downarrow$ | IntDiv $\uparrow$ |
|---|---|---|---|---|---|
| Train Set | 100% | 100% | - | 0.008 | 0.856 |
| CharRNN | 97.5% | 99.9% | 84.2% | 0.073 | 0.856 |
| VAE | 97.7% | 99.8% | 69.5% | 0.099 | 0.855 |
| JTN-VAE | 100% | 99.9% | 91.4% | 0.395 | 0.855 |
| AAE | 93.7% | 99.7% | 79.3% | 0.556 | 0.856 |
| **SiDGen** | **100%** | **88.75%** | **100%** | **0.108** | **0.893** |

### 4.2 MOLECULAR PROPERTY ANALYSIS

Figure 2 shows the distribution of key molecular properties. SiDGen generates molecules with reasonable drug-like properties: mean molecular weight of 421.57 Da, LogP of 3.07, and TPSA of 37.89 U. The maximum Tanimoto similarity to training molecules was computed for each generated molecule, with a mean value of 0.27, indicating good novelty and low memorization of training data. The synthetic accessibility score of 0.47 suggests the generated molecules are generally synthesizable.

Molecular properties were computed on canonical SMILES using RDKit descriptors (MolWt for molecular weight, LogP for LogP, TPSA for topological polar surface area). Synthetic accessibility (SA) used the Ertl SA implementation, Ertl & Schuffenhauer (2009) and reported fingerprint-based similarities use Morgan fingerprints (r=2, 2048 bits). These molecular properties are critical indicators of drug-likeness according to established medicinal chemistry guidelines. Lipinski's Rule of Five Lipinski et al. (2001) suggests that orally active drugs typically have molecular weights below 500 Da, LogP values below 5.0, and TPSA values below $140 \text{ Å}^2$ for good membrane permeability Veber et al. (2002). The synthetic accessibility score provides an estimate of synthetic feasibility, with values closer to 1 indicating easier synthesis Ertl & Schuffenhauer (2009). Low Tanimoto similarities to training data (typically below 0.4) suggest that the model generates novel chemical structures rather than memorizing training examples Reymond & Awale (2015).

### 4.3 VIRTUAL SCREENING PERFORMANCE

Virtual screening metrics were computed from ranked docking scores. The **ROC-AUC** quantifies overall ranking quality as the area under the receiver operating characteristic curve. The **enrichment factor** at $X\%$ (EF@$X\%$) is defined as

$$\text{EF@}X\% = \frac{\text{fraction of actives in top } X\%}{\text{fraction of actives expected at random}}.$$

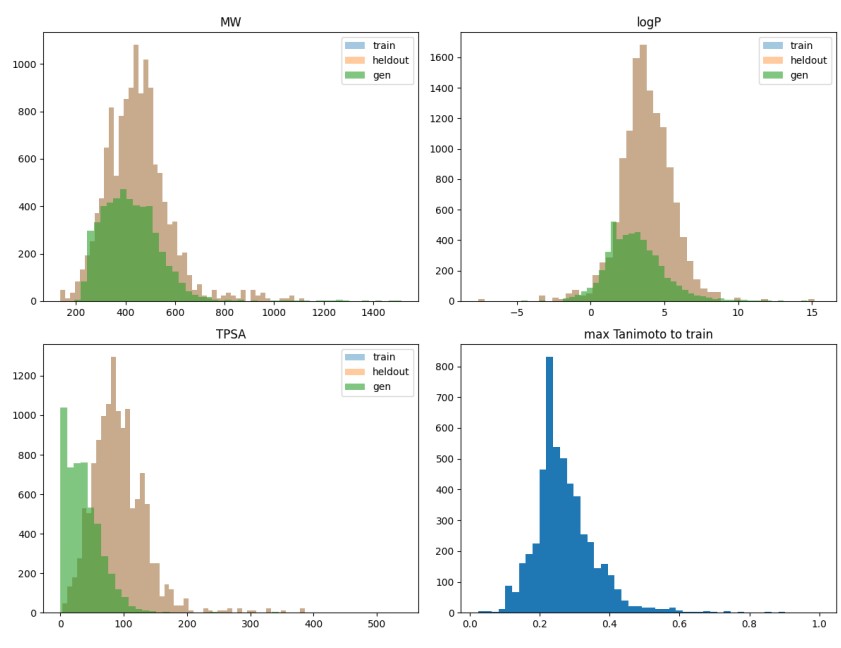

Figure 2: Properties of generated molecules.

Table 2: DUD-E Benchmark Comparison

| Method | ROC-AUC | EF@1% | BEDROC |
|--------|---------|-------|--------|
| Glide | 0.83 | – | 0.29 |
| Vina | 0.6966 | 8.82 | – |
| Gnina | 0.6799 | 7.93 | – |
| **SiDGen** | **0.819** | **10.72** | **0.28** |

**BEDROC** applies exponential early-weighting of the rankings; in our experiments we used $\alpha = 80.5$. Docking inputs, as well as active and decoy molecules, follow the DUD-E splits employed in our benchmark.

## 4.4 EXPERIMENTAL RESULTS ON CROSSDOCKED2020

**CrossDocked2020 evaluation.** The performance of SiDGen on CrossDocked2020 and selected baselines are reported in the Table 3 below.

Table 3: Performance of our model and benchmarks on CrossDocked2020.

| Model | Vina ↓ | Vina Dock ↓ | QED ↑ | SA ↑ | Diversity ↑ |
|-------|--------|-------------|-------|------|-------------|
| liGAN | - | -6.33 | 0.39 | 0.59 | 0.66 |
| Pocket2Mol | -5.14 | -7.15 | 0.56 | 0.74 | 0.69 |
| TargetDiff | -5.47 | -7.80 | 0.48 | 0.58 | 0.72 |
| DecompDiff | -5.67 | -8.39 | 0.45 | 0.61 | 0.68 |
| DecompOpt | -5.87 | -8.98 | 0.48 | 0.65 | 0.60 |
| TransDiffSBDD | -6.02 | -9.37 | 0.48 | 0.75 | 0.81 |
| PocketFlow | - | - | 0.51 | 0.29 | 0.87 |
| **SiDGen** | **-5.98** | **-9.74** | **0.55** | **0.47** | **0.89** |

## 4.5 BINDING AFFINITY PREDICTION

**PDBBind evaluation.** We report PDBBind binding-affinity prediction numbers (pKd) for SiDGen and selected baselines in Table 4 below. For PDBBind evaluation, ligands were matched by canoni-

Table 4: PDBBind binding-affinity prediction comparison (pKd).

| Paper/Model | Pearson $R$ | Spearman $\rho$ | RMSE (pKd) | MAE (pKd) |
|---|---|---|---|---|
| TargetDiff | 0.680 | 0.637 | 1.374 | 1.118 |
| EGNN | 0.648 | 0.598 | 1.445 | 1.141 |
| IGN | 0.698 | 0.641 | 1.433 | 1.169 |
| **SiDGen (this work)** | **0.695** | **0.684** | **1.075** | **0.950** |

cal SMILES, and per-SMILES predictions were aggregated by taking the most favorable (minimum) docking score when multiple poses were present. Docking free energies ($\Delta G$, in kcal·mol$^{-1}$) were converted to $pK_d$ using

$$pK_d = -\frac{\Delta G}{RT \ln 10},$$

with $T = 298$ K and $R = 1.987 \times 10^{-3}$ kcal·mol$^{-1}$K$^{-1}$. Reported Pearson $R$, Spearman $\rho$, RMSE, and MAE were computed on paired true versus predicted $pK_d$ values.

## 5 DISCUSSION

### 5.1 METHOD ANALYSIS

SiDGen successfully addresses the computational efficiency challenge in protein-conditioned molecular generation while maintaining competitive performance. The coarse-stride folding mechanism provides substantial memory and compute savings, making the approach practical for large-scale applications. The dual conditioning pathway design allows users to choose appropriate trade-offs between efficiency and structural awareness based on their specific requirements. For high-throughput screening applications, streamlined mode provides rapid generation, while full mode can be used when maximum structural fidelity is required.

### 5.2 LIMITATIONS

Several limitations should be acknowledged:

1. The coarse-stride approximation may miss fine-grained local interactions that are important for some binding modes.

2. Performance varies significantly across different protein families, suggesting the need for more robust conditioning mechanisms.

3. The model currently operates only on SMILES representations, limiting its ability to optimize 3D binding poses directly.

## 6 CONCLUSION

We present SiDGen, a structure-informed diffusion model for protein-conditioned ligand generation that addresses the critical trade-off between structural awareness and computational efficiency. Through our coarse-stride folding mechanism and dual conditioning pathways, SiDGen achieves competitive molecular generation performance while requiring significantly less computational resources than full structural conditioning approaches. Our comprehensive evaluation across multiple benchmark datasets demonstrates that SiDGen generates chemically valid, diverse molecules with reasonable drug-like properties. The variable performance across different protein targets highlights both the potential and challenges in protein-conditioned generation, providing insights for future method development. The computational efficiency gains make SiDGen particularly suitable for

high-throughput drug discovery applications where scalability is paramount. As the field moves toward larger-scale virtual screening and more complex protein targets, methods like SiDGen that balance performance with practicality will become increasingly important.

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
