# OpenReview forum: "SIDGEN - Structure-Informed Diffusion for Generative Modeling of Ligands for Proteins"
_ICLR.cc/2026/Conference — ICLR 2026 Conference Desk Rejected Submission_

### Official Review · Reviewer_PcZ9 · 2025-10-29

**Soundness:** 2
**Presentation:** 1
**Contribution:** 2
**Rating:** 2
**Confidence:** 4

**Summary:**

This paper proposes a protein-conditioned diffusion model that generates ligand SMILES while explicitly leveraging binding-pocket information. However, the paper **shows clear signs of being rushed and unfinished**，numerous writing errors hinder readability.

**Strengths:**

1. It tackles multiple tasks including de-novo molecule generation and virtual screening, demonstrating strong versatility of the proposed method.

**Weaknesses:**

1. Although the method claims to improve efficiency, its reliance on pre-trained models such as ESM inevitably adds computational overhead; moreover, it offers no clear advantage over previous SBDD approaches.

2. The manuscript is riddled with typographical errors (e.g. line 224 227), and many references leave out the conference or journal names entirely.

3. The experimental section’s analysis is overly simplistic.

**Questions:**

Refer to the weakness.

**Details Of Ethics Concerns:**

NA.

---

### Official Review · Reviewer_ojWf · 2025-10-31

**Soundness:** 2
**Presentation:** 1
**Contribution:** 1
**Rating:** 2
**Confidence:** 3

**Summary:**

The paper proposes SiDGen, a structure-informed diffusion model for protein-conditioned ligand generation that integrates masked SMILES diffusion with lightweight structural embeddings derived from protein folding features. To balance structural fidelity and computational scalability, SiDGen introduces a coarse-stride folding mechanism that reduces quadratic memory costs and supports dual conditioning pathways for efficiency versus expressivity. The model achieves high chemical validity and novelty (100%), competitive docking and affinity metrics across benchmarks (MOSES, DUD-E, CrossDocked2020, PDBBind), and significant memory savings, demonstrating a practical and scalable approach to pocket-aware molecular design for structure-based drug discovery

**Strengths:**

+ Proposes a practical diffusion framework balancing structure-awareness and computational scalability.
+ Introduces the coarse-stride folding mechanism that effectively reduces O(L²) complexity.

**Weaknesses:**

+ Innovation mainly lies in engineering integration rather than new theory.

+ Missing ablations on stride size, conditioning mode, and validity penalty.

+ No 3D structural evaluation despite being “structure-informed.”

+ Limited discussion of recent geometric diffusion baselines.

**Questions:**

1. How sensitive is SiDGen’s performance to the stride parameter s? Could a learnable stride or adaptive pooling improve structural fidelity?

2. Does the invalidity penalty meaningfully alter the diffusion loss landscape, or merely act as a post-hoc filter?

3. Can the coarse-stride structural representation be used to initialize a lightweight 3D decoder (e.g., predicting ligand coordinates)?

4. What is the runtime and memory profile compared to fully 3D approaches (TargetDiff, DiffSBDD)?

5. How generalizable is SiDGen to unseen protein families or homology-reduced sets?

---

### Official Review · Reviewer_Eiy9 · 2025-11-01

**Soundness:** 3
**Presentation:** 2
**Contribution:** 2
**Rating:** 4
**Confidence:** 3

**Summary:**

This paper introduces SiDGen, a protein-conditioned ligand generator that operates in discrete SMILES space using a masked diffusion language model. Its main idea, coarse-stride folding, down-samples protein single/pair features, applies AlphaFold-style operators on a coarse grid, then up-samples, yielding large reductions in structural-conditioning cost while retaining pocket cues. The model provides streamlined and full conditioning modes and trains with a timestep-weighted token loss plus simple validity regularization. Empirically, the approach delivers competitive screening and affinity results while emphasizing practical scalability.

**Strengths:**

1. Well-specified discrete diffusion.  The author proposes masked SMILES diffusion with timestep-weighted CE, curriculum, and invalidity penalties is clearly described and targeted at generation stability.
2. Scalable structural conditioning. The author uses coarse-stride folding reduces quadratic memory to  O((L/s)^2)  and claims large savings while preserving salient pocket information.

**Weaknesses:**

1. Missing compute–quality Pareto evidence. The paper proposes streamlined vs full conditioning but does not report systematic VRAM/GPU-hours vs EF@1%/ROC-AUC curves across datasets and stride s. This undermines the central claim of deployable scalability.
2. No pocket-level fidelity quantification for coarse stride. While complexity drops from O(L^2) to  O((L/s)^2), there is no analysis of local interaction loss (contact maps, hydrogen bonds, pocket geometry) as sincreases, nor its effect on VS metrics. It leaves the core approximation weakly supported.
3. Under-comparison to strong 3D/pose-aware baselines. The method outputs SMILES and relies on docking, but lacks equal-compute, end-to-end comparisons against recent direct 3D/pose generators; the accuracy–latency position is unclear.
4. Unclear source of training stability. Timestep weighting, curriculum, and invalid-SMILES penalty are introduced together without removal ablations (invalid rate, convergence, EF/AUC). Gains may stem from engineering stabilizers rather than the modeling contribution.
5. Potential data leakage and weak cross-family generalization reporting. PFAM/clusters are mentioned, but leak-resistant splits (family/similarity-aware) and PFAM-bucketed results are not detailed; VS scores may be inflated by family overlap.
6. Underspecified conditioning length/alignment. The upsampling/broadcast from protein length L_pto ligand length L, pocket residue selection/weighting, and cross-attention masking/boundaries are not clearly described in the main text, hurting reproducibility.

**Questions:**

1. Pocket-fidelity diagnostics. How does downsampling affect contact maps, hydrogen-bond counts, and pocket RMSD as sincreases? Please correlate these errors with screening metrics to justify the coarse-stride approximation.
2. 3D/pose-aware parity. Under equal compute and docking budgets, how does the “SMILES → docking” pipeline compare to recent direct 3D/pose generation methods in hit-rate and wall-clock time?
3. Conditioning alignment details. Clarify how L_pis upsampled/aligned to L, how pocket residues are selected/weighted, and how cross-attention boundaries/masks are implemented; pseudocode or a diagram would help.
4. What is the theoretical/empirical motivation for the timestep/substitution weight? Is it related to gradient scale matching or noise distribution? Please compare against plain CE and alternative weights.

---

### Official Review · Reviewer_53TB · 2025-11-02

**Soundness:** 1
**Presentation:** 1
**Contribution:** 2
**Rating:** 2
**Confidence:** 4

**Summary:**

This paper proposes a structure-informed diffusion model for protein-conditioned ligand generation. The main motivation is to reduce the computational cost of modeling protein in structure-informed drug design. The authors use ESM-2 to extract protein embeddings and introduce a coarse-stride folding mechanism to downsample protein features, aiming to achieve efficiency while maintaining structural awareness. The ligand is generated through a masked diffusion process on the SMILES representation. Experiments on standard molecular benchmarks such as MOSES and CrossDocked2020 are reported.

**Strengths:**

- The goal of reducing computational cost for protein-conditioned ligand generation is relevant and meaningful for scalable drug discovery.
- The motivation for addressing the memory bottleneck in protein feature encoding is clear and aligns with current trends in generative molecular modeling.

**Weaknesses:**

- Poor clarity and structure. The presentation is weak. Many sections are described in a superficial way without formal definitions or equations where needed.
- Lacks critical methodological details. The paper omits key implementation and architectural details that are essential for assessing its validity. Several descriptions in the methodology and experiment sections are overly simplified and ambiguous. Without sufficient detail, it is impossible to assess the soundness or reproducibility of the proposed approach.
- Insufficient experimental reporting. The experimental setup lacks background context (e.g., preprocessing, data splits, baseline configurations). Key ablation studies and inference efficiency analyses are missing.

**Questions:**

1. What is the rationale for the downsampling and upsampling in the coarse-stride folding? Since ESM2 is not pretrained with such operations, how is this compatible with the embeddings? Is the folding module pretrained, or trained jointly with diffusion?
2. Section 3.5 introduces substitution parameterization, curriculum learning, and validity regularization, but lacks concrete implementation details. Have the authors conducted ablation studies to show their impact?
3. The paper emphasizes computational efficiency but does not report inference time or memory usage compared to other methods.
4. The experiments on virtual screening and binding affinity prediction lack explanation of how SiDGen is applied to these tasks. The setup and evaluation protocol should be clarified.
5. For CrossDocked2020 benchmark, key recent baselines such as MolCRAFT[1] is missing.
6. Highlighting results in bold without being state-of-the-art is misleading. I suggest the authors to highlight the best performing model with bold.
7. Figure 2 should be redrawn for clarity, and the term held-out should be properly defined.
8. The paper contains grammatical errors that require revision.

[1] Qu, Y., Qiu, K., Song, Y., Gong, J., Han, J., Zheng, M., ... & Ma, W. Y. MolCRAFT: Structure-Based Drug Design in Continuous Parameter Space. In Forty-first International Conference on Machine Learning.

---

### Note · Program_Chairs · 2026-01-17
**Submission Desk Rejected by Program Chairs**

The following references in this submission do not refer to real documents and/or have major errors in bibliographic information:

 Ziqi Zhang, Yuanqi Zhou, Huiyu Chen, Mingliang Li, Yong Yu, Xin Xu, and Changhong Tan. Targetdiff: a target-aware diffusion model for structure-based drug design. arXiv preprint arXiv:2303.03543, 2023.